# Collection and Storage of Human Plasma for Measurement of Oxylipins

**DOI:** 10.3390/metabo11030137

**Published:** 2021-02-26

**Authors:** Kristen J. Polinski, Michael Armstrong, Jonathan Manke, Jennifer Seifert, Tessa Crume, Fan Yang, Michael Clare-Salzler, V. Michael Holers, Nichole Reisdorph, Jill M. Norris

**Affiliations:** 1Department of Epidemiology, Colorado School of Public Health, University of Colorado Anschutz Medical Campus, Aurora, CO 80045, USA; kristen.polinski@cuanschutz.edu (K.J.P.); TESSA.CRUME@CUANSCHUTZ.EDU (T.C.); JILL.NORRIS@CUANSCHUTZ.EDU (J.M.N.); 2Skaggs School of Pharmacy and Pharmaceutical Sciences, University of Colorado Anschutz Medical Campus, Aurora, CO 80045, USA; MICHAEL.L.ARMSTRONG@CUANSCHUTZ.EDU (M.A.); JONATHAN.MANKE@CUANSCHUTZ.EDU (J.M.); 3Division of Rheumatology, University of Colorado Anschutz Medical Campus, Aurora, CO 80045, USA; JENNIFER.SEIFERT@CUANSCHUTZ.EDU (J.S.); MICHAEL.HOLERS@CUANSCHUTZ.EDU (V.M.H.); 4Department of Biostatistics and Informatics, Colorado School of Public Health, University of Colorado Anschutz Medical Campus, Aurora, CO 80045, USA; FAN.3.YANG@CUANSCHUTZ.EDU; 5Department of Pathology, Immunology and Laboratory Medicine, University of Florida College of Medicine, Gainesville, FL 32610, USA; salzler@pathology.ufl.edu

**Keywords:** lipid mediators, oxylipins, storage, blood processing, plasma

## Abstract

Oxylipins derived from omega-3 and -6 fatty acids are actively involved in inflammatory and immune processes and play important roles in human disease. However, as the interest in oxylipins increases, questions remain regarding which molecules are detectable in plasma, the best methods of collecting samples, and if molecules are stable during collection and storage. We thereby built upon existing studies by examining the stability of an expanded panel of 90 oxylipins, including specialized pro-resolving lipid mediators (SPMs), in human plasma (*n* = 5 subjects) during sample collection, processing, and storage at −80 °C. Oxylipins were quantified using liquid chromatography-tandem mass spectrometry (LC/MS/MS). Blood samples collected in ethylenediaminetetraacetic acid (EDTA) or heparin followed by up to 2 h at room temperature prior to processing showed no significant differences in oxylipin concentrations compared to immediately processed samples, including the SPMs lipoxin A4 and resolvin D1. The majority of molecules, including SPMs, remained stable following storage for up to 1 year. However, in support of previous findings, changes were seen in a small subset of oxylipins including 12-HETE, TXB_2_, 14-HDHA, and 18-HEPE. Overall, this study showed that accurate measurements of most oxylipins can be obtained from stored EDTA or heparin plasma samples using LC/MS/MS.

## 1. Introduction

Oxylipins are endogenously generated from omega-3 and -6 polyunsaturated fatty acids (PUFAs) via enzymatic and non-enzymatic oxidation [1]. These bioactive metabolites are the mediators of PUFA effects on inflammatory and immune processes, including the resolution of inflammation and reduction in pain [2]. Though biologically potent, the relatively low abundance of oxylipins in the body poses methodological challenges in the measurement of these key metabolites. Fortunately, recent developments in targeted lipidomic methodologies have made the identification and quantification of oxylipins more reliable [3,4,5,6].

Quantifying oxylipins using lipidomics may lead to a deeper understanding of the contribution of these bioactive metabolites in human disease. For example, reductions in oxylipins known as specialized pro-resolving lipid mediators (SPMs) have been associated with airway inflammation and rheumatoid arthritis [7,8]. Longitudinal epidemiologic cohort studies with repositories of bio-specimens offer a potentially important opportunity to investigate the role of oxylipins in the development and progression of human diseases. For accurate assessments in these epidemiologic studies, though, it is crucial to minimize potential measurement errors introduced during the preanalytical phase when samples are being collected, processed, and stored. Yet, only a limited number of studies have examined how specific oxylipin concentrations are impacted by sample collection, processing, or storage conditions that samples may undergo as part of a study protocol [5,6,9,10,11,12,13]. These are summarized in Table 1.

In an epidemiologic study setting, pre-processed EDTA or heparin samples (i.e., whole blood) collected from study participants may be left at room temperature or on ice prior to processing for plasma. Early evidence for 8 oxylipins suggested that delayed plasma processing may have effects on the concentration of oxylipins such as 14,15-EpETrE [6]. Dorow et al. showed increases in 12- and 15-HETE after delayed plasma processing at room temperature [9]. Another study found that delayed sample processing at room temperature led to increases in 12-HETE, 14-HDHA, and TXB_2_ oxylipins [11]. However, that study was limited by a small sample (i.e., blood drawn from a single person on three occasions). Koch et al. conducted a comprehensive study on various additives during sample preparation, highlighting BHT use, among measurement of total oxylipins (i.e., esterified and non-esterified), and found non-significant impacts on concentrations [13]. To our knowledge, no study has examined effects on heparin plasma or a large panel of free oxylipins, including SPMs, in a delayed sample processing setting.

In addition to the delayed sample processing, plasma samples in epidemiologic cohort studies undergo long-term storage. Gladine et al. presented data on TXB_2_, 5-HETE, 12-HETE, and 15-HETE suggesting these oxylipins are stable while undergoing storage at −80 °C for two and half years [5]. Jonasdottir, et al. conducted a comprehensive study that evaluated the addition of preservatives, EDTA vs heparin, and storage on the stability of PUFAs and 18 oxylipins, and found that the area ratios of oxylipins in EDTA plasma were more stable at lower storage temperatures of −80 °C over 1 year except for increases in the oxylipins 12-HETE and TXB_2_ [10]. However, this study did not include the measurement of SPMs and utilized different preparation and analysis methods than the current study described herein. Koch et al. also found stability among total oxylipins but noted increases in 9-HETE and 8,15-DiHETE after 15 months of storage at −80 °C [13]. Furthermore, stored samples often undergo repeated freezing and thawing cycles, and limited evidence for the effects on oxylipin concentrations, especially regarding SPMs, has been presented [9]. 

The goal of this study was to identify which plasma oxylipins are suitable for investigation in epidemiologic studies based on their relative stability using a variety of metrics. Using standard epidemiologic study protocol and storage conditions, we examined the concentrations of 90 oxylipins, including SPMs, in human EDTA and heparin plasma during sample collection and storage at −80 °C over the course of a year.

## 2. Results

### 2.1. Sample Collection and Delayed Processing Experiment

EDTA or heparin plasma samples kept at room temperature for two hours showed no statistically significant differences in oxylipin concentrations by delayed processing time (Table 2 for EDTA and Table 3 for heparin). We noted heparin oxylipin concentrations tended to be greater than EDTA, particularly among arachidonic acid-derived oxylipins such as 15-HETE: 394.4 pg/mL in EDTA vs. 524.4 pg/mL in heparin. In EDTA, nominal increases in median concentrations were noted for 9,10-EpOME and 5-HEPE after 10 min at room temperature, with values about 1.7 times higher compared to baseline (9,10-EpOME: 515.8 pg/mL vs. 886.9 pg/mL; 5-HEPE: 47.6 pg/mL vs. 81.1 pg/mL); however, differences were not statistically significant. Similarly, we found the median concentration for 12-HETE in EDTA after 2 h at room temperature decreased by almost two-thirds of its concentration at baseline (491.7 pg/mL vs. 177.3 pg/mL), though not statistically significant. Of note, 9,10-EpOME, 5-HEPE, and 12-HETE had intra-day CVs greater than 25%. Regarding SPMs, lipoxin (LX) A_4_ and 15-LXA_4_ were both quantified and EDTA concentrations showed no significant change over the two hours. Resolvins were not detected in EDTA plasma.

In heparin, we found that LXA_4_ and resolvin (RV) D_1_ showed no significant changes after 1 h at room temperature. In contrast, 15-LXA_4_ concentrations after 1 h at room temperature were about 1.6 times lower compared to baseline (10.9 pg/mL vs. 6.9 pg/mL), though not statistically significant. Notably, 14-HDHA and 18-HEPE, important precursors to the SPMs RVE_1_ and maresin which were below the limit of detection in this experiment, appeared to remain stable in unprocessed heparin for 1 h at room temperature.

### 2.2. Sample Storage Experiment

Compared to baseline (i.e., fresh sample), we observed statistically significant differences in median concentrations for 8-HDoHE, 11,12-EET, 14-HDHA, 12-HETE, 8-HETE, 18-HEPE, and TXB_2_ at the one-year −80 °C storage time point (Table 4). Of note, 15-KETE, 15-LXA, and 8,9-EET were not detected at this 12-month time point, and 9-HETE, 11-HDoHE, and 15R- LXA_4_ were not detected in the storage experiment. Among oxylipins metabolized by docosahexaenoic acid (DHA), median concentrations of 8-HDOHE and 14-HDHA (precursor to maresin) nearly tripled; whereas, concentrations of 18-HEPE, a metabolite of eicosapentaenoic acid (EPA) and precursor to RVE1, significantly decreased by half at the end of the one-year storage period. The levels of oxylipins metabolized from linoleic acid (LA) (i.e., 9,10-EpOME, 9,10-DiHOME, 12,13-EpOME, 12,13-DiHOME, 9-HODE, 13-HODE, 9-OxoODE, 13-OxoODE, and EKODE), alpha-linolenic acid (ALA) (i.e., 9-HOTrE, 13-HOTrE, and 9-KOTrE), and dihomo-γ-linolenic acid DGLA (i.e., 5-, 8-, and 15-HETrE) demonstrated no significant increase or decrease at 12 months of storage. Figure 1 illustrates the changes in concentration at each storage time point compared to the fresh sample. 

The results for the spiked samples, which were prepared to have all oxylipins at detectable levels using the reference standard solution, are presented in the Appendix A. The spiked oxylipin storage pattern was identical to that of the non-spiked samples detected above. The spiked samples also provide an opportunity to examine the stability of SPMs that were not detected in the non-spiked individual samples. Generally, SPMs remained stable during long-term storage at −80 °C following biosynthesis pathway-based patterns. For example, 15-LOX and 5-LOX are key enzymes in the metabolism of 17-HDHA and downstream RVD_1_, RVD_2_, and RVD_3_. In the spiked samples, these downstream products remained stable during storage (Kruskal-Wallis *p*-values > 0.05). This was also demonstrated in most other resolvins and lipoxins on the panel with 15-LOX or 5-LOX pathways, including RVE_1_, RVD_5_, LXB_4_, and 15-LXA_4_ (Kruskal−Wallis *p*-values > 0.05). However, the following SPM concentrations did not remain stable over the measured time points: 17-RVD_1_, 10,17-DiHDoHE (protectin), 7S-Maresin, LXA_5_, and LXA_4_ (Kruskal–Wallis *p*-values < 0.05).

Table 5 summarizes the stability metrics for all 90 oxylipins considered in the present study (i.e., collection, storage at −80 °C, and variability) and highlights detectable oxylipins in EDTA from each precursor PUFA. The calibration equation average, representing the stability of the instrument and reference standard calibration curve across multiple calibrations, appeared to remain stable over the year-long experiment. Compiling findings from the two experiments, we found that the following oxylipins did not have significant differences in levels for either the sample processing or storage experiments and had CVs less than 25%: 11,12-DiHETrE, 11-HETE, 14,15-EET, 14,15-DiHETrE, 15-HETE, 5-HETE, 8,9-DiHETrE, 13-HOTrE, 9-KOTrE, 15-HETrE, 8-HETrE, 17-HDHA, 11-HEPE, 15-HEPE, 15R-LXA_4_, 17,18-DiHETE, 9,10-EpOME, 9,10-DiHOME, 12,13-EpOME, 12,13-DiHOME, 13-OxoODE. 

#### Freeze/Thaw

The concentrations of a single sample over 4 freeze-thaw cycles are presented in Figure 2 and Figure 3. When subjected to freeze-thaw cycles, LA, DGLA, and ALA-derived oxylipins appeared to remain stable over the 4 freeze-thaw cycles relative to the fresh sample concentration in this exploratory and sample limited experiment. EPA and DHA-derived oxylipins mostly appeared stable with the exception of overall decreases in 18-HEPE and increases in 14-HDHA. Among the ARA-derived oxylipins, HETEs, particularly 12-HETE and 5-HETE appeared to have increased concentrations after 2 freeze-thaw cycles. 

## 3. Discussion

In this study, EDTA and heparin samples after 2 h of delayed plasma processing at room temperature showed no significant differences in oxylipin concentrations compared to an immediately processed sample. Our findings are in support of previous studies which reported increases in 12-HETE, TXB_2_, or 14-HDHA after delayed processing of EDTA whole blood [9,11]. Notably, Dorow et al. reported no changes in these oxylipins when the samples were placed at 4 °C prior to processing [9] which may reduce ex vivo formation. Furthermore, our findings are in-line with a previous study that tested the effects of processed EDTA plasma, rather than whole blood, after 2 h at room temperature [10]. Our study adds to the existing literature by assessing SPMs, in which we found no changes in 15-LXA_4_ (in EDTA), LXA_4_ (in EDTA and heparin), or RVD_1_ (in heparin) after delayed plasma processing. In a previous study, a total 15-LXA_4_, 17-RVD_4_, and RVD_5_ were not detected [13]. Longer delayed processing times, which were not examined as part of this pilot study, may lead to significant changes. For example, Jonasdottir et al. did observe an increase in 5-HETE, 12-HETE, and TXB_2_ in processed plasma after 8 h at room temperature [10]. Of note, at each time point, we observed higher concentrations of 12-LOX products in heparin compared to EDTA plasma as follows: baseline 12-HETE at 1831.3 pg/mL in heparin vs. 491.7 pg/mL in plasma and baseline 14-HDHA at 400.3 pg/mL in heparin vs. 114.9 pg/mL in plasma. This could be explained by the expression of 12-LOX in human platelets whereby heparin is known to promote platelet activation [14]. This emphasizes the importance of reporting and harmonizing collection tube types when comparing concentrations across studies. 

Storage for up to one year at −80 °C affected only a handful of plasma oxylipins, including increases in 12-HETE, 8-HETE, TXB_2_, 14-HDHA, and 8-HDoHE, and decreases in 11,12-EET and 18-HEPE compared to baseline. As previously indicated, there is likely slow activation of residual platelets over time in plasma stored at −80 °C [10], supporting the observed increases in concentrations of 12-LOX and COX products (i.e., 12-HETE, 14-HDHA, and TXB_2_). Similarly, decreases in 11,12-EET, a CYPP450 product, may be explained by uptake through these residual platelets in stored plasma [15]. Furthermore, increases in 8-HDoHE and 8-HETE may be partially explained by their synthesis as auto-oxidation products, which can be artificially formed by in vitro auto-oxidation. The other omega-6 derived auto-oxidation products, 8-HETrE and 11-HETE, also trended in the same direction over the one-year storage period. 

Our study is unique in the assessment of a number of SPMs undergoing prolonged storage conditions. Overall, this study found that concentrations of resolvins in the spiked plasma remained largely stable. This supports our observations of resolvins in non-spiked plasma precursors (e.g., 17-HDHA) of the 5-LOX or 15-LOX pathways. Storage time passed one year, which was not examined as part of this pilot project, may result in additional differences. Previously, Jonasdottir et al. reported increases in the area ratios of 12-HETE and TXB_2_ after a one year of storage period at −80 °C, similar to our study [10]. However, our study identified additional critical oxylipins affected by storage time that have not been previously assessed. 

In an exploratory experiment, we observed that oxylipin concentrations generally remained stable during multiple freeze-thaw cycles; whereas some oxylipins, such as 12-HETE, 5-HETE, 18-HEPE, and 14-HDHA did not. This is in-line with a previous study that also noted increases in 12-HETE after two freeze-thaw cycles and increases in 5-HETE after 5 cycles [9]. Though this exploratory experiment was sample limited and conclusions on statistical significance cannot be determined, it is still advisable that freeze-thaw cycles be kept at a minimum by preparing small aliquots directly after plasma processing or keeping the number of freeze-thaw cycles consistent within the study. Additional freeze-thaw experiments with replicates are needed.

Fluctuations of oxylipin levels during processing and storage may also be due to the variability in their measurement. The 7 oxylipins that had differences in the fresh versus 12-month sample had CVs of greater than 25%; therefore, the variability over the one-year storage period may have in part been due to variability in their measurement. Overall, we did observe moderate to high intra-day variability (CVs > 20%) in about 75% of the oxylipins detected on the panel which limits our power to detect statistically significant differences. Based on the excellent technical reproducibility of the assay [16], it is likely that this variability is due to biological factors. This is an important consideration in epidemiological studies and suggests that samples should be measured at similar storage time points. Much like previous studies, our study examined non-esterified oxylipins. To our knowledge, only a couple of recent studies have reported on the effects of delayed plasma processing on total (i.e., non-esterified and esterified) oxylipin concentrations with and without additives and long-term storage [12,13]. These studies demonstrated that various storage conditions did not affect total oxylipin concentrations. Future studies are needed to replicate these findings and examine the effects of long-term storage on total SPM concentrations.

Overall, our study suggests that the concentrations of oxylipins in human plasma remain reasonably stable for up to 2 h at room temperature and up to one year of storage at −80 °C, with exceptions including the 12-LOX products and variability considerations. These findings indicate that oxylipins can be accurately quantified in epidemiologic studies using samples stored (i.e., samples not processed or analyzed immediately) for up to one year, and likely longer. 

## 4. Materials and Methods 

### 4.1. Sample Collection and Delayed Processing Experiment

To replicate conditions under which blood samples may be collected and processed during an epidemiologic study, we simultaneously collected fresh, non-fasting blood samples from 5 healthy volunteers. The age range of the participants was 27 to 66 years and all participants were non-smokers. Samples were collected via venipuncture of an arm vein into 10 mL EDTA (lavender top) and heparin tubes (green top). Plasma was prepared by centrifugation at 3000 rpm for 10 min after benchtop incubation at room temperature for 0, 10, 30, 60, and 120 min. For each time point, 120 µL aliquots of EDTA plasma were pipetted into cryovials for each subject and placed on ice prior to analysis. In parallel, a 120 µL aliquot of heparin plasma for each subject was prepared at 0 and 60 min and placed on ice prior to analysis. Figure 4 presents an illustration of the study design for this experiment as well as for the sample storage experiment described below.

### 4.2. Sample Storage Experiment

To examine the stability of oxylipin concentration measurements in fresh vs. frozen plasma, we collected fresh, non-fasting blood from 5 healthy volunteers by venipuncture of an arm vein into 10 mL EDTA tubes. EDTA plasma was promptly prepared by centrifugation at 3000 rpm for 10 min and 15,000 µL samples were processed as follows: 3 sample replicates per person were prepared for immediate quantification (i.e., fresh sample); 18 replicates per person were immediately frozen for future quantification following 3 days, 1 week, 1 month, 3 months, 6 months, and 1 year of storage at −80 °C. 

#### 4.2.1. Freeze/Thaw

Repeated freezing and thawing of samples are typical of epidemiologic studies when storage space often limits the number of aliquots. Therefore, using one participant’s ETDA plasma sample, an additional 1500 µL was prepared for the purpose of undergoing freeze-thaw cycles at the 3 days, 1 month, 6 months, and 1-year time points prior to LC/MS/MS analysis. 

#### 4.2.2. Spiked Plasma

Additionally, a 1500 µL sample was prepared by pooling EDTA plasma from all 5 participants and spiking it with 25 pg per 100 µL plasma of each oxylipin reference compound using the calibration standard solution. This pooled ETDA plasma sample was then analyzed immediately (i.e., fresh sample) and after 3 days, 6 months, and 1 year. The study was approved by the Colorado Multiple Institutional Review Board (01-675).

### 4.3. Oxylipin Sample Preparation

All standards and internal standards used for LC/MS/MS analysis of oxylipins were purchased from Cayman Chemical (Ann Arbor, Michigan, USA). All HPLC solvents and extraction solvents were HPLC grade or better. The internal standard solution (5S-HETE-d_8_, 8-ISO-PGF_2A_-D_4_, _9_)-HODE-d_4_, LTB_4_-D_4_, LTD_4_-d_5_, LTE_4_-d_5_, PGE_2_-D_4_, PGF_2a_-D_9_, Resolvin D_2_-d_5_ (RVD_2_), Resolvin D_1_-d_5_ (RVD_1_-d_5_) was formulated considering the variety of compound structures, standard availability, and reference standard retention times. 

Plasma samples were prepared using solid-phase extraction (SPE) and sample preparation generally followed a previously published method [4,17]. Briefly, proteins were precipitated from 100 µL of either heparin or EDTA plasma by adding 400 µL of ice-cold methanol and 10 µL of an internal standard solution in a 1.5 mL microfuge tube, followed by vortexing and then incubating on ice for 15 min. The samples were then centrifuged for 10 min at 4 °C at 14,000 RPM. The supernatant was transferred to a new microfuge tube and an additional 100 µL of ice-cold methanol was added to the tube and the pellet was resuspended. The samples were then placed in a microcentrifuge tube for 10 min at 4 °C at 14,000 RPM and the supernatant was removed and combined with the first supernatant. The sample was then dried in a vacuum centrifuge at 55 °C until dry. The sample was then immediately reconstituted in 1.0 mL of 90:10 water:methanol before purification by SPE. 

Oxylipins were enriched using Strata-X 33um 30 mg/mL SPE columns (Phenomenex, Torrance, CA, USA) on a Biotage positive pressure SPE manifold (Biotage, Charlotte, NC, USA). SPE Columns were pre-washed with 2 mL of MeOH followed by 2 mL of H2O. After applying the entire 1 mL of reconstituted sample, the columns were washed with 1 mL of 10% MeOH, and the oxylipins were then eluted sequentially with 1 mL of methyl formate followed by 1 mL of methanol directly into a reduced surface activity/maximum recovery glass autosampler vial (MicroSolv Technology Corp., Leland, NC, USA) drying after each solvent elution with a steady stream of nitrogen directly on the SPE manifold. The sample was then immediately reconstituted with 20 µL of ethanol and analyzed immediately.

### 4.4. Liquid Chromatography-Tandem Mass Spectrometry

Quantitation of oxylipins was performed using 2-dimensional reverse phase HPLC tandem mass spectrometry (LC/MS/MS) as previously described with some modifications [4,17]. The HPLC system consisted of an Agilent 1260 autosampler (Agilent Technologies, Santa Clara, CA, USA), an Agilent 1260 binary loading pump (pump 1), an Agilent 1260 binary analytical pump (pump 2), and a 6-port switching valve. Pump 1 buffers consisted of 0.1% formic acid in water (solvent A) and 9:1 v:v acetonitrile:water with 0.1% formic acid (solvent B). Pump 2 buffers consisted of 0.01% formic acid in water (solvent C) and 1:1 v:v acetonitrile:isopropanol (solvent D). 

10 µL of the extracted sample was injected onto an Agilent SB-C18 2.1X5mm 1.8um trapping column using pump 1 at 2 mL/min for 0.5 min with a solvent composition of 97% solvent A: 3% solvent B. At 0.51 min the switching valve changed the flow to the trapping column from pump 1 to pump 2. The flow was reversed and the trapped oxylipins were eluted onto an Agilent Eclipse Plus C-18 2.1X150 mm 1.8 um analytical column using the following gradient at a flow rate of 0.3 mL/min: hold at 75% solvent A:25% solvent D from 0–0.5 min, a linear gradient from 25–75% D over 20 min followed by an increase from 75–100% D from 20–21 min, then holding at 100% D for 2 min. During the analytical gradient, pump 1 washed the injection loop with 100% B for 22.5 min at 0.2 mL/min. Both the trapping column and the analytical column were re-equilibrated at starting conditions for 5 min before the next injection.

The mass spectrometric analysis was performed on an Agilent 6490 triple quadrupole mass spectrometer in negative ionization mode. The drying gas was 250 °C at a flow rate of 15 mL/min. The sheath gas was 350 °C at 12 mL/min. The nebulizer pressure was 35 psi. The capillary voltage was 3500 V. Data for lipid mediators was acquired in dynamic MRM mode using experimentally optimized collision energies obtained by flow injection analysis of authentic standards. Calibration standards for each oxylipin were analyzed over a range of concentrations from 0.25–250 pg on the column. Calibration curves for each lipid mediator were constructed using Agilent Masshunter Quantitative Analysis software. Plasma samples were quantitated using the calibration curves to obtain the column concentration, followed by multiplication of the results by the appropriate dilution factor to obtain the concentration in pg/mL. To ensure consistent quantitation throughout the study and standard solution stability, reference standard calibration curves were analyzed for consistency of slope.

### 4.5. Statistical Analysis

Oxylipin molecules with >20% missing observations were excluded from analyses. Concentrations are reported as picogram/mL (pg/mL). For the sample collection and processing experiment, Kruskal–Wallis *p*-values for the overall difference of oxylipin concentrations in plasma undergoing benchtop incubation were obtained. For the storage experiment, Wilcoxon rank-sum *p*-values were obtained to compare the fresh vs. 12-month time point. Additionally, for the storage experiment, fold changes were calculated as the ratio of median values to compare oxylipin concentrations at each storage time point with the fresh sample. To visualize these changes, we constructed a heat map using heatmap.2 in R Studio 3.4. Coefficients of variation (CV) were calculated from the 3 sample replicates of fresh plasma samples obtained in the storage experiment. Lastly, to show the plasma concentrations of each oxylipin measured at baseline and following each of the four freeze-thaw cycles, line graphs were constructed using ggplot2 in R Studio 3.4. 

## Figures and Tables

**Figure 1 metabolites-11-00137-f001:**
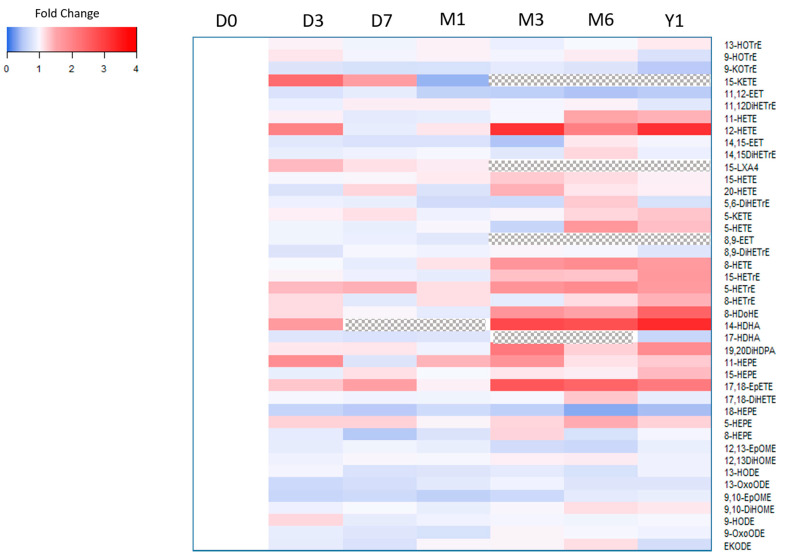
Storage time experiment. Heatmap depicting the fold change of oxylipin concentrations. Blue color indicates lower concentration compare to fresh sample, whereas red color indicates higher concentration. A dotted pattern indicates oxylipin was not detected. Abbreviations: Day 0 (D0), Day 3 (D3), Day 7 (D7), Month 1 (M1), Month 3 (M3), Month 6 (M6), Year 1 (Y1).

**Figure 2 metabolites-11-00137-f002:**
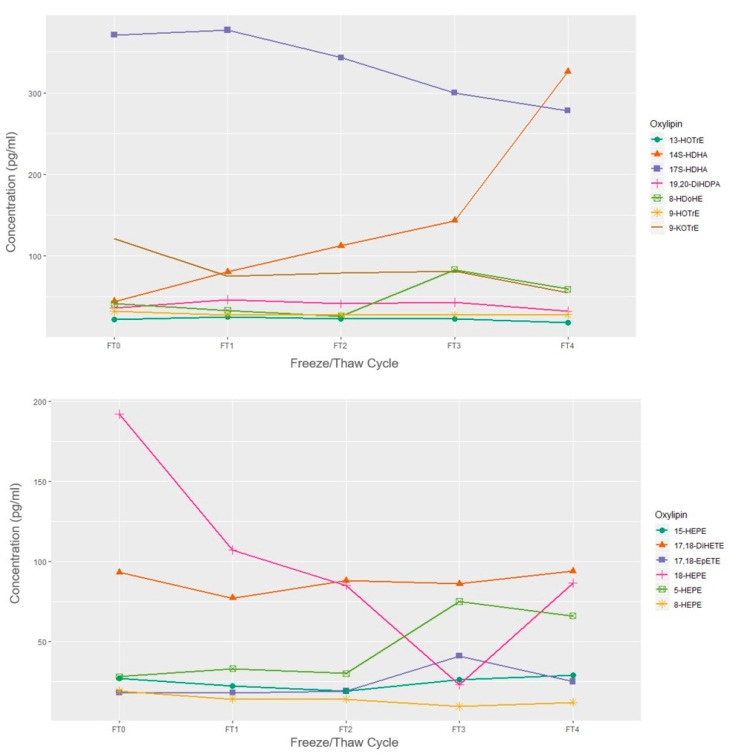
Concentrations of omega-3 derived oxylipins by freeze/thaw cycle based on *n* = 1 EDTA plasma sample. FT0 indicates fresh sample; FT1 indicates 1 freeze/thaw cycle; FT2 indicates 2 freeze/thaw cycles, etc.

**Figure 3 metabolites-11-00137-f003:**
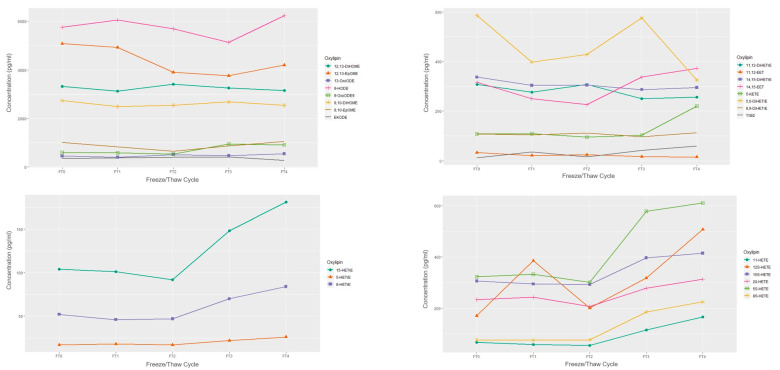
Concentrations of omega-6 derived oxylipins by freeze/thaw cycle based on *n* = 1 EDTA plasma sample. FT0 indicates fresh sample; FT1 indicates 1 freeze/thaw cycle; FT2 indicates 2 freeze/thaw cycles, etc.

**Figure 4 metabolites-11-00137-f004:**
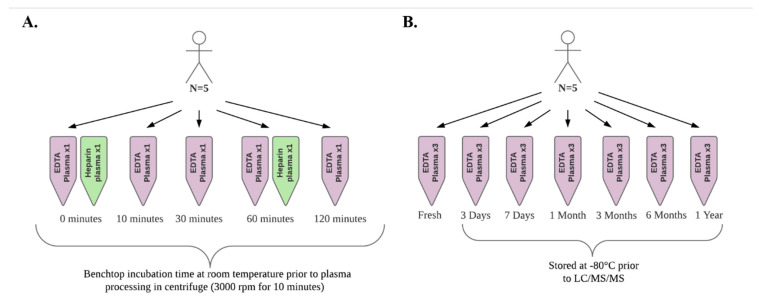
Illustration of the sample collection and processing experiment (**A**) and the sample storage experiment (**B**) study designs.

**Table 1 metabolites-11-00137-t001:** Summary of existing studies reporting on the effects of storage or delayed plasma processing conditions on oxylipin concentrations (including the present study).

Author, Year	*n*	Quantification of Oxylipins (n) ^1^	Storage Conditions			Delayed Plasma Processing Conditions			Freeze/Thaw Cycles
			Temperature	Duration	Sample Type(s)	Temperature	Duration	Sample Type(s)	
Present Study	5	non-esterified plasma oxylipins (46)	−80 °C	3 days, 7 days, 1 month, 3 months, 6 months, and 1 year	EDTA plasma (no additives)	Room Temperature	0, 10, 30, 60, and 120 min	pre-processed EDTA and heparin plasma (i.e., whole blood)	up to 4 cycles
Willenberg 2015 [6]	1	non-esterified plasma oxylipins (8)				Room temperature or ice	5, 30, 60, and 120 min	pre-processed EDTA plasma (i.e., whole blood)	
Dorow, 2016 [9]	6	non-esterified plasma and serum oxylipins (28)	4 °C or room temperature	30, 60, 90, 120 min	EDTA plasma (with and without additives)	4 ℃ or room temperature	30, 60, 90, and 120 min	pre-processed EDTA plasma (i.e., whole blood)	up to 5 cycles
			−20 °C, −80 °C, and −150 °C	1 day, 1, 2, and 6 months					
Jonasdottir, 2018 [10]	4	non-esterified plasma oxylipins (18)	Room temperature	2 and 8 h	EDTA and heparin plasma (with and without additives)				
			6 °C	2, 8, and 24 h					
			−20 °C	24 h and 1 week or 1, 4, 12, 26, and 52 weeks					
			−80 °C	4, 12, 26, and 52 weeks					
Ramsden, 2019 [11]	1 (on three separate occasions)	non-esterified plasma oxylipins (16)				Room Temperature or ice	0, 10, 20, 30, 60, and 120 min	pre-processed EDTA plasma (i.e., whole blood)	
Gladine, 2019 [5]	5	non-esterified plasma oxylipins (4)	−80 °C	2.5 years	EDTA plasma (no additives)	
Rund, 2020 [12]	6	non-esterified and total plasma and serum oxylipins (23)				4 ℃ ^2^	4 h, 24 h, or after pneumatic tube system transport	pre-processed EDTA plasma and serum (i.e., whole blood) ^2^	
Koch, 2020 [13]	4	non-esterified and total plasma oxylipins (133)	−80 °C	1−6, 9, 12, and 15 months	EDTA plasma	4 ℃	24 h	pre-processed EDTA plasma (i.e., whole blood)	
						20 ℃	4 h	processed EDTA plasma	
						−20 ℃	5 days	(with and without additives)	

^1^ number in parentheses indicates the number of oxylipins on panel; ^2^ whole blood was at room temperature for 30 min prior to storage at 4 °C before the plasma processing.

**Table 2 metabolites-11-00137-t002:** Concentrations (pg/mL) of oxylipins in EDTA plasma at each time point in the sample processing experiment (*n* = 5).

Oxylipin	T0	T10	T30	T60	T120	*p*-Value *
11,12-EET	20.1 (11.4)	26 (26.7)	14.9 (9.8)	17.7 (3.9)	21.9 (15.5)	0.92
11,12-DiHETrE	222.4 (81.9)	243.3 (10)	226.3 (18.4)	237.5 (39.8)	250.4 (27.2)	0.74
11-HETE	103.0 (46)	93.6 (27.3)	92.7 (46.7)	94.4 (62.6)	90.9 (57.6)	0.99
12-HETE	491.7 (155.4)	326.9 (28.7)	281 (73.9)	249.9 (152.9)	177.3 (198.1)	0.32
14,15-EET	259.3 (100.1)	218.6 (31.4)	219.9 (33.3)	273.1 (11.5)	252.4 (27.5)	0.65
14,15-DiHETrE	199.0 (61)	217.1 (24.9)	206.5 (18.5)	215.8 (39.8)	224.9 (44.5)	0.86
15-LXA_4_	11.5 (4.1)	8.7 (1.8)	8.95 (1.35)	7.1 (0.6)	8.5 (3.3)	0.37
15-HETE	394.4 (119.9)	370.5 (246.3)	431.1 (203.7)	373.6 (135.1)	422.1 (247.3)	0.98
20-HETE	152.8 (76)	175.4 (18.2)	209.4 (110.8)	239.6 (80)	220.4 (148.2)	0.85
5,6-DiHETrE	618.7 (177.1)	740 (126.6)	625.4 (196)	641.4 (314.6)	676.5 (38.9)	0.90
5-KETE	120.8 (54.8)	114 (31.7)	66.2 (21.3)	100.9 (23.2)	62 (34.1)	0.54
5-HETE	396.3 (230.6)	307.2 (38.5)	307.2 (23.2)	336.7 (83.8)	343.2 (83.8)	0.88
8,9-DiHETrE	115.6 (32.6)	129.3 (19.8)	129.6 (32.8)	114.3 (34)	136.1 (43.6)	0.73
8-HETE	135.2 (30.3)	135.8 (22.7)	107.7 (13.5)	129 (27.9)	131.3 (35.6)	0.85
9-HETE	91.2 (19.4)	70.6 (45.3)	103.2 (39)	82.2 (46.6)	109.7 (90.3)	0.92
LXA_4_	41.6 (11.6)	31.4 (3.2)	26.9 (4.6)	31.3 (10.4)	32 (7.2)	0.16
PGJ_2_	22.3 (6)	16.5 (1.4)	19.6 (4.1)	17.9 (11)	19.6 (11.7)	0.94
12,13-EPOME	3240.8 (2539)	4255.7 (3280.1)	2422.2 (3977.1)	3827.6 (2512.7)	3860.2 (1480.3)	0.91
12,13-DiHOME	3928.0 (2801.4)	3670.7 (2915.3)	3316.1 (3834.4)	3861.5 (3385.7)	3792 (3574.4)	1.00
13-HODE	35,458 (29.457)	43,784 (52,306)	29,965 (52,477)	30,773(50,510)	36,191 (39,458)	1.00
13-OxoODE	492.7 (447)	646 (135.1)	555.8 (353.5)	451.2 (259.4)	690.6 (191.6)	0.82
9,10-EPOME	515.8 (547.4)	886.9 (636.4)	427.7 (853.1)	516.8 (549.4)	891.6 (578.4)	0.93
9,10-DiHOME	2393.7 (1901.4)	2463.7 (1799.4)	2093.1 (1969.4)	2379.6 (1748)	2413 (1708.4)	1.00
9-HODE	4218.5 (1796.3)	5098.4 (4391.3)	3795.1 (5510.5)	3748.4 (3828.9)	3901.7 (5749.5)	1.00
9-OxoODE	1011.0 (399)	1033 (238.7)	756.2 (206.3)	885.2 (250.7)	895.3 (369.1)	0.70
EKODE	429.1 (373.1)	512.4 (448.8)	395.7 (179.7)	323.1 (413)	406.5 (297.2)	0.77
15-HETrE	145.8 (86.3)	142.2 (43.1)	142.4 (48.2)	135.3 (60.6)	179.5 (76)	0.99
5-HETrE	42.5 (11.5)	41.8 (10.7)	38.7 (19.6)	35.7 (13.5)	42.1(19.8)	0.99
8-HETrE	35.3 (54.4)	37.3 (33.3)	41.6 (37.9)	37.9 (38.3)	34 (45.9)	0.90
13-HOTrE	23.4 (8)	25.2 (16.3)	21.2 (13.2)	27.9 (15.5)	24.3 (17.8)	0.99
9-HOTrE	22.6 (29.1)	18.3 (18.7)	17 (32.7)	20.2 (29.6)	24.6 (33)	0.92
9-KOTrE	46.0 (49.9)	67.8 (11.9)	61.6 (13.2)	63.9 (61.6)	59.05 (41.9)	0.99
11-HDoHE	106.3 (43.6)	83.5 (88)	84.3 (62.7)	62.3 (29.8)	50.1 (8.3)	0.07
14-HDHA	114.9 (42.9)	105.4 (99.4)	84.85 (30.3)	103.1 (54.8)	90.9 (42.1)	0.67
17-HDHA	447.8 (200.7)	392.8 (143.8)	378 (155.6)	431.8 (172.9)	427.6 (93.2)	0.75
19,20-DiHDPA	48.4 (33)	54 (12.4)	48.7 (27.4)	34.8 (28.2)	55.3 (51)	0.70
8-HDoHE	81.6 (23.5)	57.2 (36.6)	79.2 (19.3)	72.2 (38.2)	84 (7.8)	0.83
15-HEPE	37.3 (5.1)	39.1 (4.3)	32.5 (17.1)	48.1 (17.6)	37.8 (13.9)	0.89
17,18-EpETE	62.1 (11)	60.7 (28.8)	63.5 (15.5)	63 (27.8)	56.9 (14.8)	0.98
17,18-DiHETE	70.2 (43.4)	83.3 (35.1)	68.7 (16.4)	84.5 (43.6)	72.7 (36.7)	0.95
18-HEPE	189.2 (40.8)	191.9 (22.2)	162 (54.7)	182.6 (40.6)	164.2 (41.8)	0.86
5-HEPE	47.6 (21.2)	81.1 (36)	72.2 (32.6)	51.4 (37.5)	73.8 (31.1)	0.96
8-HEPE	25.0 (21.3)	30.8 (17.5)	33.7 (11.1)	23 (26.3)	27.5 (1.6)	0.77

All values presented as median (interquartile range). * Kruskall−Wallis *p*-value for the overall difference of oxylipin concentrations in EDTA plasma that remained on the benchtop at room temperature for 0, 10, 30, 60, and 120 min.

**Table 3 metabolites-11-00137-t003:** Concentrations (pg/mL) of oxylipins in Heparin plasma at each time point in the sample processing experiment (*n* = 5).

Oxylipin	T0	T60	*p*-Value *	Oxylipin	T0	T60	*p*-Value *
13-HOTrE	18 (8.5)	20.7 (9.4)	0.60	5-HETrE	53.7 (18.8)	49.2 (21.9)	0.92
9-HOTrE	20.7 (9.4)	14.9 (19.8)	0.60	8-HETrE	65.8 (34)	56.4 (37.4)	0.92
11-HDoHE	472.2 (387.7)	579.5 (334.5)	0.92	11,12-EET	44.5 (14.8)	40.8 (22.5)	0.75
14-HDHA	400.3 (297.6)	483.5 (292.8)	0.92	11,12-DiHETrE	215.6 (46.5)	214.5 (17.9)	0.92
17-HDHA	426.3 (135.9)	429.8 (77.1)	0.92	11-HETE	161.9 (21.4	125.1 (29.8)	0.17
19,20-DiHDPA	30.5 (17.6)	27.7 (7.3)	0.83	12-HETE	1831.3 (502.7)	1521.5 (148.8)	0.25
8-HDoHE	121.9 (63.5)	137.4 (9.6)	0.46	12-HHTrE	223.6 (285.3)	224 (374.3)	0.62
RVD_1_	3.7 (5.3)	3.9 (6.9)	0.71	14,15-EET	800.7 (142.7)	710.1 (80.9)	0.25
15-HEPE	30.8 (30)	30.2 (17)	0.46	14,15-DiHETrE	217.8 (69.9)	203.9 (77.3)	0.75
17,18-EpETE	112.6 (78.5)	101.9 (29.2)	0.92	15-LXA_4_	10.9 (3.6)	6.9 (0.7)	0.05
17,18-DiHETE	52 (18.3)	47.9 (30)	0.92	15-HETE	524.4 (144.2)	446.4 (53)	0.46
18-HEPE	205.7 (14.7)	205 (42.6)	0.92	20-HETE	116.2 (57.4)	145.7 (156.6)	0.35
5-HEPE	141.3 (41.6)	140 (18)	0.46	5,6-DiHETrE	1222.9 (301.4)	1073.8 (151.6)	0.75
8-HEPE	30.7 (6.9)	29.3 (7.8)	0.92	5-KETE	216.7 (37.3)	136.9 (36.7)	0.35
12,13-EPOME	3720.8 (2129.7)	3034.4 (2417.7)	0.75	5-HETE	1107.2 (305.1)	1004.6 (216.6)	0.46
12,13-DiHOME	3244.4 (1975.2)	3419.2 (2042)	0.92	8,9-DiHETrE	125.9 (32.3)	133 (34.5)	0.25
13-HODE	35,908.6 (3337.9)	32,104.2 (35,691.9)	0.75	8-HETE	161.9 (9)	150.3 (24.5)	0.75
13-OxoODE	346.8 (138.1)	389.9 (94.5)	0.60	9-HETE	109.3 (33.8)	104.7 (2.7)	0.92
9,10-EPOME	478.3 (326.6)	447.5 (389.2)	0.35	LXA_4_	32 (16.7)	24.7 (1.8)	0.60
9,10-DiHOME	1915.6 (1331.8)	2001.8 (1535.1)	0.92	LTB_4_	11.6 (17.7)	34.8 (16.8)	0.08
9-HODE	4387.7 (1927.6)	3572.2 (1589.7)	0.60	PGJ_2_	19.4 (5.1)	16.8 (2.7)	0.75
9-OxoODE	940.9 (339)	648 (158.8)	0.17	TXB_2_	88.7 (84.9)	45.5 (49.5)	0.60
EKODE	306.7 (99)	166.7 (69.4)	0.25	9-HEPE	31.1 (23.8)	44.1 (17.2)	0.58

All values presented as median (interquartile range).* Kruskall−Wallis *p*-value for the overall difference of oxylipin concentrations in Heparin plasma that underwent benchtop incubation at room temperature for 0 and 60 min Oxylipins not reported in Table 3 include those that were not detected in any subjects: 10,17-DiHDoHE, 11-PGF_2α_, 11-dehydro-TBX, 12-HEPE, 12-KETE, 14,15-EpETE, 14,15-DiHETE, 15-KETE, 17-RVD1, 20-hydroxy-LTB4, 5, 15-DiHETE, 6-α-PG, 6-kete-PGF_1α_, 7s-Maresin, 8,9-EET, 8,15-DiHETE, 8-iso-15-PGF_2α_, 8-iso-PGF_2α_, dinor-11B-PGF_2α_, dinor-6-keto-PGF_2α_, dinor-8-iso-PGF_2α_, LXB_4_, LTD_4_, LXA_5_, PGB_2_, PGD_2_, PGE_2_, tetranor-PGFM, RVD_2_, RVD_3_, RVD_5_, and RVE_1_ or were detected in at least 1 subject but not all: 15-deoxy-Δ12,14-PGJ_2_, 19,20-EpDPE, 5,6-EpETrE, 6-trans-LTB_4_, 7r-Maresin, TXA_2_, LTE_4_, PGF_2α_, tetranor-PGEM.

**Table 4 metabolites-11-00137-t004:** Concentrations (pg/mL) of oxylipins in EDTA plasma at each time point in the storage experiment (*n* = 5).

Oxylipin	Fresh	Day 3	Day 7	Month 1	Month 3	Month 6	Month 12	*p*-Value *
8-HDoHE	29 (14)	34 (12)	29.5 (14)	25 (12)	51 (9)	45 (15)	72 (29)	**0.03**
12-HHTrE	58 (39)	278 (309)	100.75 (111)	59 (85)	359.5 (462)	502 (581)	293 (255)	0.17
14-HDHA	50 (22.5)	84 (26)			143 (25)	138 (32)	164.5 (60)	**0.04**
15-KETE	120.5 (63.5)	285 (122)	195 (52)	46 (26)				NA
11-HEPE	7 (0)	13 (6)	5.5 (2.5)	10 (3.5)	12.5 (1)	8 (2)	9 (8)	0.76
15-HEPE	27 (17)	24 (15)	31 (12)	27 (22)	30 (8)	29 (11)	37 (13)	0.68
TXB_2_	12 (2)	90.5 (148.5)	17.5 (12.75)	19 (10)	49 (58)	46 (81)	87.25 (64)	**0.04**
11,12-EET	30 (10)	23 (12)	26 (5)	18 (1)	17 (11)	14 (3)	16 (3)	**0.03**
11,12-DiHETrE	234 (51)	212 (79)	251 (90)	251 (83)	226 (62)	243 (23)	193 (56)	0.42
11-HETE	60 (23)	63 (15)	51 (8)	53 (17)	59 (10)	92 (46)	86 (8)	0.09
12,13-EpOME	5082 (1910)	4390 (1038)	4806 (2415)	4499 (1814)	3526 (1163)	3326 (81)	4573 (1033)	0.83
12,13-DiHOME	2574 (2310)	2390 (2479)	2589 (2180)	2546 (2276)	2724 (2448)	2787 (2155)	2403 (2655)	0.83
12-HETE	196 (20)	398 (143)	172 (20)	219 (47)	623 (283)	409 (54)	627 (258)	**0.03**
13-HODE	39,497 (24,524)	38,047 (21,533)	30,455 (22,657)	31,351 (27,704)	33,346 (21,538)	29,678 (15,572)	36,535 (27,286)	0.99
13-HOTrE	22 (4)	23 (5)	21 (4)	23 (3)	20 (13)	22 (17)	24 (11)	0.76
13-OxoODE	452 (181)	301 (17)	329 (39)	378 (116)	423 (114)	361 (86)	361 (223)	0.99
14,15-EET	305 (56)	254 (52)	234 (92)	235 (44)	144 (27)	337 (159)	292 (45)	0.68
14,15-DiHETrE	229 (43)	203 (93)	213 (90)	227 (111)	190 (123)	276 (104)	208 (69)	0.37
15-LXA4	13 (3)	18 (3)	15 (7)	14 (4)	.	.	.	NA
15-HETrE	104 (46)	107 (39)	96 (46)	90 (39)	140 (48)	138 (75)	177 (48)	0.09
15-HETE	316 (53)	310 (47)	321 (43)	346 (57)	406 (51)	370 (22)	332 (22)	0.99
17,18-EpETE	20 (18)	26 (10)	32 (4)	21 (9)	53 (11)	49 (66)	43 (12)	0.32
17-HDHA	371 (72)	299 (82)	285 (75)	295 (77)	.	.	235 (82)	0.06
17,18-DiHETE	93 (30)	91 (32)	87 (27)	88 (25)	92 (9)	121 (51)	82 (5)	0.48
18-HEPE	266 (31)	165 (80)	141 (33)	182 (80)	149.5 (83)	88 (29)	116 (18.5)	**0.03**
19,20-DiHDPA	33 (13)	36.5 (37.5)	37 (17)	31 (18)	72 (113)	41 (27)	62 (147)	0.4
20-HETE	172 (121)	134 (64)	208 (110)	133 (137)	249 (76)	192 (119)	183 (110)	0.99
5,6-DiHETrE	586 (57)	540 (205)	520 (153)	409 (125)	400 (115)	754 (582)	436 (91)	0.24
5-HEPE	38 (21)	47 (3)	47 (18)	39 (11)	46 (31)	56 (29)	47 (3)	0.68
5-HETrE	18 (32)	25 (25)	26 (12)	21 (19)	32 (7)	34 (19)	30 (21)	0.99
5-KETE	108 (15)	115 (38)	124 (28)	100 (45)	110 (31)	131 (38)	142 (36)	0.06
5-HETE	336 (36)	320 (39)	300 (82)	338 (48)	211 (51)	582 (292)	454 (92)	0.32
8,9-EET	732 (835)	695 (688)	663 (927)	606 (1313)	.	.	.	NA
8,9-DiHETrE	108 (26)	85 (34)	106 (27)	101 (21)	110 (31)	107 (34)	87 (15)	0.32
8-HEPE	18 (32)	15.5 (2)	9 (1.5)	14 (13)	22 (6)	13.5 (6)	17.5 (19.5)	0.9
8-HETrE	44 (17)	52 (11)	37 (16)	51 (4)	38 (11)	52 (23)	63 (7)	0.17
8-HETE	68 19)	68 (13)	60 (8)	77 (25)	119 (32)	127 (42)	114 (44)	**0.03**
9,10-EpOME	852 (480)	542 (375)	577 (438)	495 (418)	566 (352)	727 (463)	761 (380)	0.68
9,10-DiHOME	1719 (1681)	1578 (1584)	1720 (1275)	1532 (1582)	1734 (1784)	1997 (1228)	1884 (2036)	0.99
9-HODE	4073 (1849)	4912 (1390)	3553 (1839)	3814 (1856)	3937 (1474)	3858 (1546)	4005 (1986)	0.99
9-HOTrE	25 (10)	28 (3)	24 (5)	26 (3)	24 (3)	27 (3)	19 (8)	0.68
9-KOTrE	84 (25)	66 (13)	62 (20)	63 (12)	71 (7)	66.5 (14.5)	44 (25)	0.09
9-OxoODE	600 (157)	526 (119)	485 (56)	446 (148)	620 (256)	593 (228)	564 (326)	0.99
EKODE	360 (220)	313 (177)	275 (20)	367 (84)	371 (146)	417 (323)	260 (108)	0.32

All values presented as median (interquartile range). * two-sample Wilcoxon rank-sum *p*-value comparing fresh vs. frozen (month 12); Bold numbers indicate statistical significance.

**Table 5 metabolites-11-00137-t005:** Oxylipins by precursor PUFA summarized by stability metrics in EDTA plasma.

Oxylipin	Biosynthesis Pathway	Calibration Equation Average ^1^		Sample Processing *p*-Values (Table 2) ^2^	Sample Storage *p*-Values (Table 3) ^3^	Intraday CVs ^4^
		Average Slope	CV	T0, 10, 30, 60, 120 min	Fresh vs. Frozen	Fresh EDTA Plasma
ALA						
13-HOTrE	15-LOX	4.51	11.3%	0.99	0.76	17.1%
9-HOTrE	NA	22.40	7.4%	0.92	0.68	28.2%
9-KOTrE	NA	0.60	13.9%	0.99	0.09	14.1%
EPA						
11-HEPE ^5^	ROS	1.08	7.0%	NA	0.76	13.7%
12-HEPE ^6^	12-LOX	0.08	28.2%	NA	NA	NA
14(15)-EpETE ^6^	CYP	0.49	11.4%	NA	NA	NA
14,15-DiHETE ^6^	EH	0.05	5.3%	NA	NA	NA
15-HEPE	15-LOX	0.78	5.5%	NA	0.68	17.2%
17(18)-EpETE	CYP	0.93	8.2%	NA	0.32	34.1%
17,18-DiHETE	EH	3.23	4.1%	NA	0.48	14.0%
18-HEPE	15-LOX	0.61	19.5%	NA	0.03	25.9%
5-HEPE	5-LOX	1.01	8.9%	NA	0.68	36.4%
8-HEPE	ROS	0.78	7.5%	NA	0.90	29.3%
9-HEPE^6^	ROS	0.37	7.5%	NA	NA	NA
LXA_5_ ^6^	5-LOX	2.44	2.3%	NA	NA	NA
RVD_5_ ^6^	15-LOX	0.79	1.7%	NA	NA	NA
RVE_1_ ^5^	CYP/5LOX	1.81	5.2%	NA	NA	NA
DHA						
10,17-DiHDoHE ^6^	15-LOX	1.62	1.7%	NA	NA	NA
11-HDoHE ^5^	ROS	0.90	7.9%	0.07	NA	NA
14-HDHA	12-LOX	1.50	10.3%	0.67	0.04	60.7%
17-HDHA	15-LOX	0.43	15.0%	0.75	0.06	13.3%
17R-RVD_1_ ^6^	15-LOX	0.60	4.6%	NA	NA	NA
RVD_1_ ^6^	15-LOX	1.22	6.4%	NA	NA	NA
RVD_2_ ^6^	15-LOX	0.40	6.8%	NA	NA	NA
RVD_3_ ^5^	15-LOX	3.07	2.3%	NA	NA	NA
19,20-EpDPE ^5^	CYP	0.71	10.1%	NA	NA	NA
19,20-DiHDPA	EH	7.00	17.0%	NA	0.40	21.8%
7R Maresin-1 ^5^	12-LOX	1.00	4.4%	NA	NA	NA
7S Maresin-1 ^6^	12-LOX	0.26	6.8%	NA	NA	NA
8-HDoHE	ROS	0.80	8.3%	NA	0.03	29.1%
LA						
12(13)-EpOME	CYP	0.34	20.0%	0.91	0.83	15.7%
12,13-DiHOME	EH	2.20	7.7%	1.00	0.83	14.5%
13-HODE	15-LOX	0.14	7.7%	1.00	0.99	21.4%
13-OxoODE	15-LOX	0.56	4.3%	0.82	0.99	14.0%
9(10)-EpOME	CYP	2.09	14.0%	0.93	0.68	18.9%
9,10-DiHOME	EH	3.35	4.4%	1.00	0.99	14.9%
9-HODE	NA	1.66	10.3%	1.00	0.99	21.1%
9-OxoODE	NA	0.50	2.6%	0.70	0.99	20.7%
EKODE	NA	2.32	6.7%	0.77	0.32	21.3%
DGLA						
15-HETrE	15-LOX	3.24	8.6%	0.99	0.09	16.3%
5-HETrE	5-LOX	3.03	6.8%	0.99	0.99	20.7%
8-HETrE	ROS	1.25	8.7%	0.9	0.17	16.7%
ARA						
11(12)-EET	CYP	0.90	8.0%	0.92	0.03	16.9%
11,12-DiHETrE	EH	1.97	3.8%	0.74	0.42	8.2%
11-HETE	ROS	7.58	8.3%	0.99	0.09	11.2%
11B-PGF_2a_ ^5^	COX	21.6	5.5%	NA	NA	NA
11-dehydro-TBX_3_ ^6^	COX	7.3	6.4%	NA	NA	NA
12-HHTrE	COX	0.31	11.1%	NA	0.17	29.2%
12-HETE	12-LOX	0.64	10.7%	0.32	0.03	17.7%
12-KETE ^6^	12-LOX	0.07	19.4%	NA	NA	NA
15-KETE ^6^	15-LOX	1.5	12.2%	NA	NA	NA
14(15)-EET	CYP	5.93	13.8%	0.65	0.68	15.2%
14,15-DiHETrE	EH	3.01	3.6%	0.86	0.37	9.9%
15R-LXA_4_ ^6^	5-LOX	1.19	5.5%	0.37	NA	14.5%
15-HETE	15-LOX	1.05	8.6%	0.98	0.99	11.2%
15-deoxy-Δ12,14-PGJ_2_ ^5^	COX	3.86	5.4%	NA	NA	NA
20-HETE	CYP	0.57	6.0%	0.85	0.99	21.6%
20-hydroxy-LTB_4_ ^6^	5-LOX	3.94	6.5%	NA	NA	NA
5,6-EpETrE ^5^	CYP	0.03	12.7%	NA	NA	NA
5,6-DiHETrE	EH	1.15	8.0%	0.9	0.24	20.8%
5-KETE	5-LOX	0.62	7.6%	0.54	0.06	24.9%
5-HETE	5-LOX	1.26	5.0%	0.88	0.32	14.7%
5,15-DiHETE ^5^	5-LOX	0.8	2.4%	NA	NA	NA
6-α-PG ^6^	COX	11.4	4.5%	NA	NA	NA
6-keto-PGF_1α_ ^6^	COX	3.81	4.2%	NA	NA	NA
6-trans-LTB_4_ ^6^	5-LOX	2.05	61.1%	NA	NA	NA
8(9)-EET	CYP	0.69	30.0%	NA	NA	19.2%
8,9-DiHETrE	EH	0.93	5.0%	0.73	0.32	16.3%
8,15-DiHETE ^5^	5-LOX	0.63	5.0%	NA	NA	NA
8-iso-15-PGF_2α_ ^5^	ROS	8.67	25.3%	NA	NA	NA
8-iso-PGF_2α_ ^5^	ROS	14.82	5.9%	NA	NA	NA
8-HETE	ROS	0.97	11.4%	0.85	0.03	20.2%
9-HETE	ROS	1.03	6.7%	0.92	NA	NA
TXA_2_ ^5^	COX	0.59	14.1%	NA	NA	NA
TXB_2_	COX	12.62	4.7%	NA	0.04	42.2%
dinor-11B-PGF_2α_ ^6^	COX	2.68	4.1%	NA	NA	NA
dinor-6-keto- PGF_2α_ ^6^	COX	1.03	6.3%	NA	NA	NA
dinor-8-iso- PGF_2α_ ^6^	COX	5.36	4.9%	NA	NA	NA
LXA_4_ ^6^	5-LOX	0.73	14.8%	NA	NA	NA
LXB_4_ ^6^	5-LOX	0.31	5.4%	NA	NA	NA
LTB_4_ ^6^	5-LOX	2.83	3.4%	NA	NA	NA
LTD_4_ ^6^	5-LOX/GST	2.59	36.6%	NA	NA	NA
LTE_4_ ^6^	5-LOX/GST	2.45	43.4%	NA	NA	NA
PGB_2_ ^6^	COX	0.98	4.1%	NA	NA	NA
PGE_2_ ^6^	COX	1.28	3.4%	NA	NA	NA
PGF_2α_ ^5^	COX	40.23	5.8%	NA	NA	NA
PGJ_2_ ^5^	COX	2.74	5.3%	NA	NA	NA
PGD_2_ ^6^	COX	4.73	4.3%	NA	NA	NA
tetranor-PGEM ^5^	COX	4.20	11.2%	NA	NA	NA
tetranor-PGFM ^6^	COX	4.61	11.9%	NA	NA	NA

^1^ Represents the stability of the instrument and references standard calibration curve across multiple calibrations. ^2^ Kruskall−Wallis *p*-value for the overall difference of oxylipin concentrations in EDTA plasma that underwent benchtop incubation at room temperature for 0, 10, 30, 60, and 120 min. ^3^ Two-sample Wilcoxon rank-sum *p*-value comparing fresh vs. frozen (month 12). ^4^ Coefficients of variation (CV)s for oxylipins measured in fresh plasma. The CVs were calculated for each oxylipin measured in 3 replicates of the same fresh plasma samples. ^5^ Oxylipin detected at greater than LOD but less than LOQ in at least 1 subject at a one-time point in the EDTA sample storage experiment. ^6^ Not detected in EDTA plasma. NA represents “not applicable”.

## Data Availability

The data presented in this study are available on request from the corresponding author.

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
