# Peer review of "Collection and Storage of Human Plasma for Measurement of Oxylipins"

_metabolites, 2021, doi:10.3390/metabo11030137_

Round 1

Reviewer 1 Report

The authors demonstrated for the majority of oxylipins the stability concerning different handling and storage conditions, which is of importance for application in clinical research. Unfortunately, the results of freeze/thaw experiment are not significant due to the low number of replicates (n=1) and the authors ignore current and quite similar work published by the group of Hils-Helge Schebb (published in Talanta, 2020, 121074).

Minor remarks:

  1. Nomenclature: Numbers within oxylipins names have to be subscripted like prostaglandin E2. The same is true for the numbers in the deuterated standards (line 356 following)
  2. Line 116 What is the EDTA concentration?
  3. Table 2 and 3: are the p-values from the comparision of T0 vs the other timepoints? Then more p-values should be availbale. Please clarify.
  4. Values of tables: I would prefer using mean values and the standard deviation, which is more common for analytic and better for small number of replicates.
  5. Long term experiments: 9-HETE is missing here. Why is the oxylipin not shown here? You detected it as shown in table 2 and 3. This is an important non-enzymatic build lipid mediator and their profile could be very helpful to understand peroxidation processes.
  6. Did you compare the results from EDTA and heparin plasma?

Major remarks:

  1. Freeze/thaw experiments: This is an interesting experiment and the results would be very helpful for sample handling. But the authors only used one replicate, so the results can´t be significant. I suggest to delete this part.
  2. The authors mentioned the results as unique and did not take a current paper of the Schebb group (Stability of oxylipins during plasma generation and long-term storage, Talanta, 2020) into account. Within this paper 113 oxylipins were analyzed including a long term experiment (12 and 15 month) and time span before centrifugation with quite similar time points than in your paper. They show differences in particular for 9-HETE and 13-HODE during long term storage. The difference to the present work is, that the use an alkaline hydrolysis step to analyze the total amount. Please comment on this.

Reviewer 2 Report

This is a well written manuscript with a well grounded theory

Author Response

We thank the reviewer for taking the time to review this work and for their supportive feedback. 

Reviewer 3 Report

Polinski et al. provide valuable insights in the stability of Oxylipins in plasma. These mediators play vital roles in pro- and anti-inflammatory signaling and have gained increasing interest throughout the years. The study at hand is straight forward and very relevant to the field. The authors give considerable evidence on the stability of Oxylipins and also point out limitations in the analytical value of samples that have been stored over longer time frames. Polinski et al. show for the first time that Oxylipins might not only be degraded over time but, in some cases, even display elevated levels at some observed time points. The latter is problematic in future studies and needs to be pointed out. The study is well conducted and thoroughly designed. The reviewer is a little disappointed by the limited power of the evidence but is aware of the workload that is associated with the experiments. I have no further critical remarks and endorse the study for publication.

Author Response

We thank the reviewer for their time and careful review. 

Round 2

Reviewer 1 Report

The present revision significantly improved over the original version and the authors have addressed all my concerns and remarks.